# Uncoupling Protein 3 Catalyzes the Exchange of C4 Metabolites Similar to UCP2

**DOI:** 10.3390/biom14010021

**Published:** 2023-12-22

**Authors:** Jürgen Kreiter, Tatyana Tyschuk, Elena E. Pohl

**Affiliations:** Institute of Physiology, Pathophysiology and Biophysics, University of Veterinary Medicine, 1210 Vienna, Austria; jkreiter@stanford.edu (J.K.); tatyana.tyschuk@proton.me (T.T.)

**Keywords:** SLC25 protein family, substrate transport, aspartate, malonate, malate, sulfate

## Abstract

Uncoupling protein 3 (UCP3) belongs to the mitochondrial carrier protein superfamily SLC25 and is abundant in brown adipose tissue (BAT), the heart, and muscles. The expression of UCP3 in tissues mainly dependent on fatty acid oxidation suggests its involvement in cellular metabolism and has drawn attention to its possible transport function beyond the transport of protons in the presence of fatty acids. Based on the high homology between UCP2 and UCP3, we hypothesized that UCP3 transports C4 metabolites similar to UCP2. To test this, we measured the transport of substrates against phosphate (^32^P_i_) in proteoliposomes reconstituted with recombinant murine UCP3 (mUCP3). We found that mUCP3 mainly transports aspartate and sulfate but also malate, malonate, oxaloacetate, and succinate. The transport rates calculated from the exchange of ^32^Pi against extraliposomal aspartate and sulfate were 23.9 ± 5.8 and 17.5 ± 5.1 µmol/min/mg, respectively. Using site-directed mutagenesis, we revealed that mutation of R84 resulted in impaired aspartate/phosphate exchange, demonstrating its critical role in substrate transport. The difference in substrate preference between mUCP2 and mUCP3 may be explained by their different tissue expression patterns and biological functions in these tissues.

## 1. Introduction

Mitochondrial uncoupling proteins (UCPs) are a subfamily of membrane protein homologues belonging to the solute carrier superfamily SLC25 [1,2,3,4,5,6]. UCP1 is the most prominent subfamily member abundant in brown adipose tissue (BAT), where it mediates non-shivering thermogenesis under cold-acclimated conditions [7,8,9,10,11]. UCP1 catalyzes fatty acid (FA)-dependent proton transport [12,13,14,15,16] across the inner mitochondrial membrane (IMM). It leads to the uncoupling of the transmembrane proton gradient, generated by oxidative phosphorylation, from ATP production and the generation of heat.

After being discovered in 1997 [17,18], UCP2 and UCP3 were identified as potential uncouplers due to their sequence homology with UCP1 and their ability to decrease the mitochondrial membrane potential and increase membrane conductance for protons [19,20]. The latter was demonstrated in experiments on proteoliposomes [21,22] or lipid bilayer membranes [23,24] reconstituted with the recombinant UCP2 and UCP3. Their role in non-shivering thermogenesis has been discounted primarily due to the low protein amount in tissues [25]. Instead, it has been suggested that UCP2 and UCP3 could regulate proton leakage to reduce the formation of reactive oxygen species (ROS) [26,27].

The expression of UCP2 and UCP3 has been linked to a specific cell metabolism type [28,29]. UCP2 was found in cells and tissues that mainly rely on aerobic glycolysis, such as immune, stem, and cancer cells [30,31,32]. These cell types were shown to predominantly convert glucose to lactate even in the presence of oxygen (Warburg effect). In addition to proton transport in the presence of FA [23], UCP2 catalyzes the exchange of various C4 metabolites for phosphate plus a proton, as demonstrated in cell and artificial systems [33,34]. UCP3 is abundant in BAT, heart, and skeletal muscle [28,35,36,37], tissues that predominantly rely on FA oxidation. With emerging evidence for substrate transport catalyzed by other UCPs [38,39], the focus ought to fall on UCP3 as the closest homologue of UCP2. UCP3 shares 72% of sequence homology with human and 73% with mouse UCP2 [28]. The homology is particularly high in the putative substrate translocation pathway [40] (Figure 1A). It is especially striking when compared to another member of the SLC25 superfamily with dual function, the adenine nucleotide translocase (ANT1, Figure 1B). In addition to its nucleotide transport function, a proton transport function in the presence of long-chain fatty acids (FA) has been proposed for ANT1 [41,42,43]. According to the proposed transport mechanism, FA anion, nucleotides (ATP, ADP), and substrate transport inhibitors (carboxyatractyloside and bongkrekic acid) compete for the same binding site, R79 [44].

In the present study, we hypothesized that UCP3 may catalyze an efficient exchange of various C4 metabolites against radioactively labeled phosphate similar to UCP2, but with a difference in substrate transport preferences resulting from their cell metabolism-dependent tissue expression. In addition, the critical binding site for the substrate and FA anion may be the same as recently shown for ANT1 [44]. To test these hypotheses, we reconstituted a purified recombinant mouse UCP3 (mUCP3) and its mutant mUCP3-R79S into liposomes and measured the exchange rates of different C4 metabolites against radiolabeled phosphate.

## 2. Materials and Methods

### 2.1. Chemicals

Unless otherwise mentioned, all chemicals were purchased in highest purity from Sigma Aldrich GmbH (Vienna, Austria), Merck Millipore (Darmstadt, Germany), or Carl Roth GmbH (Karlsruhe, Germany). Lipids 1,2-dioleoyl-sn-glycero-3-phosphocholine (DOPC), 1,2-dioleoyl-sn-glycero-3-phosphoethanolamine (DOPE), and cardiolipin (CL) were from Avanti Polar Lipids Inc. (Alabaster, AL, USA). Radioactive substrates [2,5′,8-^3^H]-ATP, ^14^C-L-malic acid, and ^32^P-phosphate were obtained from Perkin Elmer (Waltham, MA, USA).

### 2.2. Homology Modeling of Mouse UCP3 and Homology Score Calculation

A homology model of mUCP3 was made using the SWISS-MODEL Workspace/GMQE [45]. The tertiary structure of the protein was computed using the amino acid sequence of mUCP3 (Uniprot #: P56501) and the crystallographic structure of the bovine ADP/ATP carrier (PDB: 1OKC, [46]). Sequence homology scores between mUCP3 and mUCP2 (Uniprot #: P70406) and mANT1 (Uniprot #: P48962) were calculated in 10 amino acid increments from the sequence alignments (Appendix A), using the Sim Alignment Tool (https://web.expasy.org/sim/) (accessed on 1 February 2023) and colored as depicted in Figure 1.

### 2.3. Production and Reconstitution of Recombinant Proteins into Liposomes

Expression, induction, inclusion, body isolation, and reconstitution into the liposomes of mUCP3, mUCP2, and mANT1 were performed as described elsewhere [24,47,48]. In brief, the open reading frame of mUCP3, mUCP2, or ANT1 was added to the E. coli expression strain Rosetta. Inclusion bodies containing the target protein were extracted by cell disruption using a French press and resuspended into TE/G buffer (100 mM Tris at pH 7.5, 5 mM EDTA, and 10% *v*/*v* glycerin) containing 2% *w*/*v* lauroylsarcosine. For protein reconstitution, 1 mg of protein from inclusion bodies was solubilized in TE/G-buffer containing 2% SLS and 1 mM DTT and mixed gradually with 50 mg of a lipid mixture (DOPC:DOPE:CL—45:45:10 mol%) dissolved in TE/G-buffer with the addition of 1.3% Triton X-114, 0.3% octylpolyoxyethylene, 1 mM DTT, and 2 mM GTP. After overnight incubation, the mixture was concentrated using Amicon^®^ Ultra-15 filters (Millipore, Schwalbach, Germany), dialyzed for 2 h against 1 L of TE/G-buffer containing 1 mg/mL BSA and 1 mM DTT, and then dialyzed twice overnight against 1 L of TE/G without DTT. Buffer exchange was performed by three dialyses against 1 L of assay buffer (50 mM Na_2_SO_4_, 10 mM MES, 10 mM Tris, and 0.6 mM EGTA at pH = 7.35). Aggregated and unfolded proteins were removed by centrifugation of the dialysate at 14,000× *g* and application of a 0.5 g hydroxyapatite column (Bio-Rad, Munich, Germany). GTP, initially added for protein stabilization, was diluted 7-fold, so that the amount of GTP in the proteoliposomes was negligible.

The remaining detergent was removed by incubating twice with Bio-Beads SM-2 (Bio-Rad, Germany). The protein content of proteoliposomes was determined by a Micro-BCA Protein Assay (Perbio Science Deutschland GmbH, Bonn, Germany). The protein purity was verified by silver staining (Appendix A). Proteoliposomes were stored at −80 °C until used, but not longer than two months.

### 2.4. Formation of Unilamellar Liposomes

DOPE, DOPC, and CL lipids were mixed in chloroform at a 45:45:10 mol% concentration and evaporated under nitrogen flow until a fine film was formed on the wall of the glass vial. To ensure full evaporation of chloroform, the vial was connected to a vacuum pump for 30 min. Assay buffer (50 mM Na_2_SO_4_, 10 mM TRIS, 10 mM MES, 0.6 mM EGTA; pH = 7.34) was added to the lipids, and the solution was vortexed to dissolve the lipids. Lipids were mixed with mUCP2-, or mUCP3- or ANT1-containing liposomes to reach a final lipid concentration of 4 mg/mL and a protein-to-lipid ratio of 4 µg/(mg of lipid). For empty liposomes, the lipid mixture was dissolved at 4 mg/mL with assay buffer. Liposomes were then incubated with 2 mM of phosphate (Pi) or malate and 10 to 50 µL ^32^P-Pi or 10 µL ^14^C-malate as tracer. Unilamellar liposomes were formed with an extruder (Avanti Polar Lipids, Inc., Alabaster, AL, USA) using a 400 nm polycarbonate pore filter, followed by a membrane filter with a 100 nm pore diameter. Unincorporated substrates were removed via size-exclusion chromatography. The homogeneity of the liposomal size was continuously verified by dynamic light scattering (Zetasizer, Malvern Panalytical Ltd., Malvern, UK).

### 2.5. Radioactive Transport Assay

The release of intraliposomal substrates was initiated by adding 2 mM of the chosen non-radioactive substrate to the extraliposomal volume. Substrate exchange was stopped at t = 10 s, 60 s, and 600 s by adding the sample to 30 mL of exclusion resin. Initial radioactivity of the (proteo-)liposomes was assessed by adding the sample to the resin before adding extraliposomal substrates. For each time point, the first 20 mL from the flow-through was collected in 4 × 5 mL fractions to (i) measure the peak signal, corresponding to the remaining amount of radioactive substrate in the (proteo-)liposomes, and (ii) collect the background signal as a quality control. Peak radioactivity was usually found in fractions 2 and 3, while the background signal was obtained in fractions 1 and 4. To each fraction, 10 mL of scintillation cocktail was added, and radioactivity was measured by liquid scintillation counting (Tri-Carb 2100TR, Perkin Elmer). Inhibitors were added prior to liposome formation and were present intraliposomal and in the bulk solution to account for the random orientation of UCP3 in the membrane. 

### 2.6. Determination of Exchange Rates

Data are represented by the remaining radioactive substrate, which is the counts per minute (CPM) of the radioactive substrates over time divided by the CPM from ***t*** = 0 s. The data were then fitted to an exponential function, where ***y***_0_, ***a***, and ***b*** are the fitting parameters.
(1)$$y\left(t\right)=y_{0}+a\cdot e^{-b\cdot t}$$
(2)$$y_{0}+a=1$$

Equation (2) is required to meet the starting condition. The exchange rate ***k*** is then derived from the initial slope of the exponential function, where **[*X*_0_]** = 2 mM is the starting concentration of the respective radioactive substrate: (3)$$k=-\left[X_{0}\right]\frac{\partial }{\partial t}y\left(t\right)=\left[X_{0}\right]\cdot a\cdot b\;$$

### 2.7. Site-Directed Mutagenesis

In vitro site-directed mutagenesis was carried out on expression plasmids containing the cDNA of mUCP3 as templates. The mutation was introduced with designed oligonucleotides to alter mUCP3 Arg84 (CGC) to Gln (CAG) (R84Q) using a QuikChange II site-directed mutagenesis kit (Agilent, Vienna, Austria). Successful mutations of the expression plasmids were confirmed by sequencing. Mutant mUCP3 expression plasmids were transformed into the E. coli expression strain Rosetta. Expression, induction, inclusion, body isolation, and reconstitution into the liposomes of the mUCP3 mutant were performed as described above. 

### 2.8. Docking of Phosphate and Malate to UCP3

The structure of UCP3 was prepared for docking by the removal of non-standard residues and non-complexed ions, the addition of hydrogen, and the assignment of Gasteiger charges in UCSF Chimera [49]. The ligand structures (phosphate and malate) were retrieved from the ZINC database (https://zinc.docking.org/, accessed last on 27 February 2023). Static docking was performed using AutoDock [50,51], embedded in UCSF Chimera. The grid box was set around the putative binding site. The docking was performed in 10 iterations per substrate, with one iteration picked based on binding energy and distance between arginine residues and the substrate molecule. The docking image was visualized in PyMOL (PyMOL Molecular Graphics System, Version 2.0, Schrödinger, LLC, Portland, OR, USA).

### 2.9. Statistical Analysis

The data are presented as the mean and standard deviation of at least three independent experiments.

## 3. Results

### 3.1. mUCP3 Exchanges Malate Against Phosphate

First, we investigated whether mUCP3 reconstituted into liposomes transports radioactively labeled ^14^C-malate against phosphate, using mUCP2 as a positive control [33] and ANT1 as a negative control (Figure 2A). Empty liposomes, in which the exchange should be driven by diffusion over the membrane only, were used as an additional negative control (Figure 2A). By comparing the efflux of ^14^C-malate from proteoliposomes measured over time, we concluded that mUCP3 catalyzes the malate/phosphate exchange similarly to mUCP2 (Figure 2B). While ANT1 catalyzes the exchange of ATP against ADP (Appendix A), it did not induce any significant malate/phosphate exchange, as expected from its tight substrate specificity [52,53]. Thus, we conclude that both mUCP3 and mUCP2 are active transporters of malate and phosphate. We calculated exchange rates of 9.57 ± 4.39 µM/s (corresponding to 12.0 ± 5.5 mmol/min/g_protein_) and 14.5 ± 3.5 µM/s (18.1 ± 4.4 mmol/min/g_protein_) for mUCP3 and mUCP2, while the rates for empty and ANT1-containing liposomes were 0.19 ± 0.01 µM/s and 0.23 ± 0.09 µM/s, respectively (Figure 2C and Table 1). We also tested whether UCP3-mediated phosphate/malate exchange is inhibited by GTP, GDP, and the substrate analogue phenylsuccinate (PS). We measured the amount of ^32^P-phosphate remaining in the liposomes after 600 s compared to the initial amount at the beginning of the experiment. We found inhibition values of 76.3 ± 18.2% in the presence of 10 mM GTP, 47.0 ± 2.0% for 10 mM GDP, and 92.2 ± 4.6% for 20 mM PS (Figure 2D and Appendix A).

### 3.2. The Homoexchange of Malate and Phosphate by mUCP3

Next, we examined the homoexchange of malate and phosphate mediated by mUCP3. We filled mUCP3-containing liposomes either with ^14^C-malate or ^32^P-phosphate and measured their release against malate or phosphate, respectively (Inserts in Figure 3A,B). For comparison, we measured these exchange rates for mUCP2 as well. We found that the malate exchange is higher for mUCP2 (16.6 ± 3.5 µM/s) than for mUCP3 (7.94 ± 1.88 µM/s), while the phosphate exchange is higher for mUCP3 (11.7 ± 3.2 µM/s) compared to mUCP2 (8.00 ± 1.40 µM/s) (Figure 3C and Table 1). 

### 3.3. mUCP3 Catalyses the Heteroexchange of Phosphate against Various Substrates

In the next step, we analyzed mUCP3-mediated heteroexchange of phosphate against a selection of mitochondrial relevant substrates that were tested for UCP2. We filled UCP3-containing liposomes with ^32^P-phosphate and measured the release of phosphate (Figure 4A and Appendix A). Since the proteins are randomly inserted into the liposomes (approximately 50:50), we can only globally determine the transport rate but not its directionality. We consider this by assuming that only half of the proteins are involved in the specific exchange. If the exchange of substrates is independent of the protein orientation, the exchange rates determined here would be half the value. We found that the exchange of phosphate against aspartate (19.1 ± 4.6 µM/s), sulfate (14.0 ± 4.1 µM/s), and sulfite (12.9 ± 4.5 µM/s) is higher than against malate (Figure 2). Lower transport rates were observed for oxaloacetate (5.07 ± 1.92 µM/s), thiosulfate (2.70 ± 0.36 µM/s), malonate (1.96 ± 0.72 µM/s), succinate (1.16 ± 0.17 µM/s), and citrate (1.10 ± 0.39 µM/s). In contrast, little to no transport was measured for asparagine (0.40 ± 0.04 µM/s), oxoglutarate (0.37 ± 0.04 µM/s), glutamate (0.29 ± 0.07 µM/s), and glutamine (0.27 ± 0.12 µM/s). All values are also shown in mmol/min/g protein in Table 1. Here, we decided to take twice the “apparent” malate/phosphate exchange rate of empty liposomes (Figure 2) to define non-transported substrates. We tested the same set of substrates on the mUCP2-mediated exchange with phosphate and found strong differences with mUCP3 (Figure 4B and Appendix A). In addition, malate, the highest exchange rates were determined for sulfate (18.4 ± 3.7 µM/s) and malonate (16.5 ± 2.0 µM/s). Much lower exchange rates were for aspartate (1.31 ± 0.19 µM/s), oxaloacetate (1.26 ± 0.18 µM/s), and citrate (1.14 ± 0.38 µM/s). Very low to no exchange was measured for thiosulfate (0.54 ± 0.36 µM/s), succinate (0.52 ± 0.19 µM/s), glutamate (0.36 ± 0.06 µM/s), oxoglutarate (0.32 ± 0.04 µM/s), asparagine (0.27 ± 0.04 µM/s), sulfite (0.24 ± 0.06 µM/s), and glutamine (0.24 ± 0.02 µM/s) (see also Table 1). While most of the substrates are not transported at all or only at a very slow rate (Figure 4C, white bars), aspartate, oxaloacetate, succinate, sulfite, and thiosulfate have a significantly higher exchange rate for mUCP3 compared to mUCP2, with a ratio of exchange rate (k_UCP3_/k_UCP2_) of 14.6 ± 4.1, 4.0 ± 1.6, 2.23 ± 0.88, 53.8 ± 23.0, and 5.0 ± 3.4, respectively (Figure 4C, green bars). In contrast, malate and malonate are transported faster by mUCP2, with a ratio of the exchange rates (k_ucp3_/k_ucp2_) of 0.66 ± 0.31 and 0.12 ± 0.05, respectively (Figure 4C, orange bars). For citrate (0.96 ± 0.47) and sulfate (0.76 ± 0.27), the exchange rate is almost equal for both proteins (Figure 4C, grey bars). In the absence of any external substrate, we did not observe any significant ^32^P-phosphate efflux, ruling out phosphate uniport for UCP3 and UCP2 (“None” in Figure 4 and Appendix A). 

### 3.4. Arginine 84 (R84) of UCP3 Plays a Crucial Role in mUCP3-Mediated Substrate Exchange

Finally, we performed static docking experiments to identify most of the existing phosphate binding sites by using the homology model structure of mUCP3 (Figure 1). Since the aspartate/phosphate exchange had the highest exchange rate, we also tested the docking of aspartate to mUCP3. We found a high probability for phosphate to bind to R84 and R184 (Figure 5A) and for aspartate to bind to R84 and R278 (Figure 5B). Since R84 appeared in both docking results, we produced the mUCP3R84Q mutant and examined the effect of the mutation on the aspartate/phosphate heteroexchange (Figure 5C). Indeed, the exchange rate was almost 40-fold lower for the R84Q mutant (0.50 ± 0.08 µM/s, Figure 5D) and comparable with non-transported substrates (Figure 4A, Table 1). Thus, R84 of mUCP3 has a significant impact on the UCP3-mediated heteroexchange of aspartate against phosphate. 

## 4. Discussion

Our present results show that mouse UCP3 facilitates membrane transport of negatively charged metabolites, which otherwise have a rather low intrinsic membrane permeability [54]. The exchange rate decreases in the following order: aspartate > sulfate ≈ sulfite ≈ phosphate ≈ malate >> oxaloacetate >> thiosulfate ≈ malonate ≈ succinate ≈ citrate. These results are similar to those obtained for recombinant human UCP3 in parallel with our study [55]. As mUCP2 and mUCP3 are highly homologous at the substrate binding site (Figure 1), we expected similarities in their substrate specificity. However, there are clear differences between the two proteins: mUCP3 showed the highest exchange rates for aspartate, sulfate, and sulfite, whereas mUCP2 transported sulfate and malonate at the highest rate. Because the tissue expression of mUCP2 and mUCP3 is different [28], we suggest that the difference in substrate affinity may be explained by differences in the main tissue metabolism. For example, the export of aspartate catalyzed by UCP3 from mitochondria may play a role in attenuating the development of hypertrophy in cardiomyocytes under stress conditions when cellular metabolism shifts toward glycolysis [56]. By transporting sulfate, UCP3 could contribute to cardioprotection [57]. The highly membrane-permeable signaling molecule H_2_S [58] is degraded to membrane-impermeable sulfite in the mitochondrial matrix, and sulfite is in turn degraded to sulfate by sulfite oxidase, which is in the intermembrane space. As UCP3 also exhibits a high transport rate for sulfate, it may help to reduce ROS damage in the heart and skeletal muscle by exporting sulfate from the mitochondria, encouraging further sulfite/sulfate exchange. Human UCP5 and UCP6, which are less homologous to UCP3 than UCP2, have previously been proposed to contribute to ROS regulation by exporting H_2_S degradation products, sulfite and thiosulfate, from mitochondria in exchange for sulfate [57]. Thus, it remains an open question whether UCP3 transports sulfoxides. Interestingly, hUCP5 and hUCP6 are also proposed to transport dicarboxylates and aspartates, although to a lesser extent. In addition, the insect UCP4 from *Drosophila melanogaster* was reported to catalyze the homoexchange of aspartate and, with much lower activity, the uniport transport of aspartate [38]. However, uniport transport was not observed for mammalian UCP2, UCP5, and UCP6 [33,57]. Thus, UCP3-mediated uniport transport of aspartate is not supported by the much closer mammalian homologs.

Both mUCP3 and mUCP2 catalyze the homoexchange of phosphate and malate, but with different kinetics. While the homoexchange of mUCP3 is faster for phosphate (compared to mUCP2), the homoexchange of malate is faster for mUCP2 (compared to mUCP3). Substrate homoexchange seems strange at first, but several mitochondrial carriers have been shown to mediate homoexchange, such as the ADP/ATP carrier (ADP and ATP transport in both directions) and the phosphate carrier. Both transport modes are correlated with the metabolic needs of the mitochondria and whether the ATP synthase produces ATP from ADP and phosphate or breaks up ATP into ADP and Pi. The exchange rates determined in this study are of the same order of magnitude for the different mitochondrial carriers and their respective substrates (Appendix A). The Pi/Pi homoexchange rate (in mmol/min/g protein) is slightly higher for the phosphate carrier [59] than for UCP3 (PiC: 70–90; UCP3: 11.7 ± 3.2). In contrast, the malate/malate homoexchange mediated by the citrate carrier (CiC) is lower [60] compared to UCP3 (CiC: 1.98 ± 0.16; UCP3: 7.94 ± 1.88).

The calculated transport rates of mUCP2 in this study are comparable to those reported for other uncoupling proteins [33,38,39]. We generally found a similar substrate specificity for mUCP2, except for citrate, which was reported to not be transported by hUCP2 [33]. The transport rates of malate, malonate, and sulfate were 3- to 6-fold higher in our study, which could be due to the higher concentration of substrates used (2 mM compared to 1 mM [33]. In contrast, the transport rates of aspartate and oxaloacetate are lower [33]. However, it should be emphasized that exchange rates in different studies can be better compared by the determination of the apparent binding constant K_m_ and the maximum exchange rate V_max_ according to the Michaelis–Menten relation. With the apparent K_m_ of 1 mM for phosphate [24] and 2.4 ± 0.1 mM for malate [38], the transport rates determined in this paper are expected to be higher than those determined by Vozza et al. Another possible explanation for some of the differences in the results could be that Vozza et al. investigated human UCP2 by measuring ^32^Pi influx, whereas in the present study, mouse UCP2 was investigated by measuring ^32^Pi efflux.

Meanwhile, other orthologous uncoupling proteins have been shown to be substrate carriers. Plant UCP1 and UCP2 were functionally characterized as amino acid/dicarboxylate transporters [39], while UCP4A from *Drosophila melanogaster* catalyzes the unidirectional transport of aspartate [38]. UCP4 from *Caenorhabditis elegans* was reported to transport succinate, which controls complex II-mediated oxidative phosphorylation [61]. In contrast, several mitochondrial substrate transporters from the SLC25 group, such as ATP/ADP carrier (ANT), phosphate carrier (PiC), dicarboxylate carrier (DIC), and oxoglutarate carrier (OGC), have been shown to have protonophoric activity as part of a dual transport function.

The importance of the central arginine ring for the binding of ATP, ADP, and fatty acid anion has already been demonstrated for ANT, UCP1, and UCP3 [24,41,44,62,63]. We have hypothesized that the substrate binds in the same region of mUCP3. Indeed, the mutation of R84 resulted in impaired aspartate/phosphate exchange, demonstrating its critical role in substrate transport, and suggesting a mechanism similar to that of ANT1 [44]. The latter involves at least one of the arginines (R79) in the central binding site in the substrate and FA anion transport pathways.

Our results shed light on the possible dual transport function that UCP3 may play in the mitochondrial metabolism of the tissues in which it is expressed. They also highlight the importance of further investigation of the possible substrate transport function of members of the uncoupling protein family.

## Figures and Tables

**Figure 1 biomolecules-14-00021-f001:**
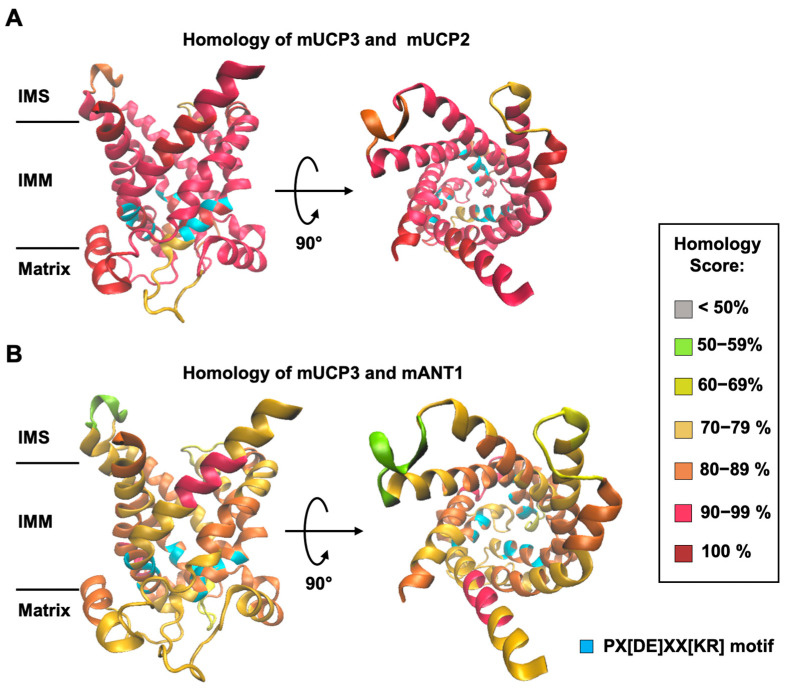
Homology of mUCP3 to mUCP2 and mANT1 mapped on a structural model of mUCP3. A cartoon representation of the structural model of UCP3 derived via mapping over the crystallographic structure of ANT1 (PDB code: 1OKC) The homology score between mUCP2 and mUCP3 (**A**) and mANT1 and mUCP3 (**B**) was assessed at 10 amino acid intervals. Structures are shown from the side (left) and top (right). Abbreviations are IMS—intermembrane space; IMM—inner mitochondrial membrane. The blue color denotes the mitochondrial carrier motif PX[DE]XX[KR], which is present in all mitochondrial carriers of the SLC25 family.

**Figure 2 biomolecules-14-00021-f002:**
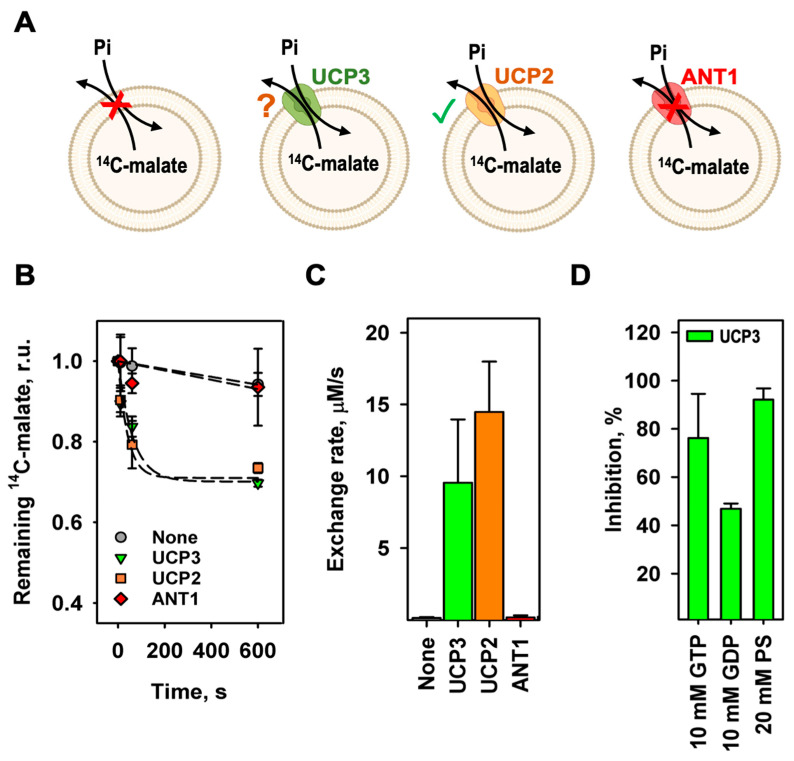
The malate/phosphate exchange is mediated by mUCP3 and mUCP2. (**A**) Experimental setup to test the mUCP3-mediated malate/phosphate exchange. (**B**) Efflux of malate from liposomes over time measured by remaining intraliposomal ^14^C-malate radioactivity. Lines are a least-squares fit of an exponential function to data. Concentrations of intraliposomal malate and extraliposomal phosphate were 2 mM. Liposomes were measured without protein (grey circles) with reconstituted mUCP3 (green triangles), mUCP2 (orange squares), or ANT1 (red diamonds). (**C**) Exchange rate calculated for the malate/phosphate exchange of empty liposomes (first bar) and liposomes containing mUCP3 (second bar), mUCP2 (third bar), or ANT1 (fourth bar). (**D**) Inhibition of phosphate/malate exchange by GTP, GDP, and phenylsuccinate (PS). Liposomes were filled with 2 mM ^32^P-phosphate. The extraliposomal substrate was malate at a concentration of 2 mM. Inhibitors were present in both the liposomes and the bulk solution and were added before the start of the exchange. Membranes were made of 45:45:10 mol% PC:PE:CL. The buffer solution contained 50 mM Na_2_SO_4_, 10 mM Tris, 10 mM MES, and 0.6 mM EGTA at pH = 7.34 and T = 296 K. Lipid concentration was 4 mg/mL; protein concentration was 4 µg/(mg of lipid). The data are the mean ± SD of at least three independent experiments.

**Figure 3 biomolecules-14-00021-f003:**
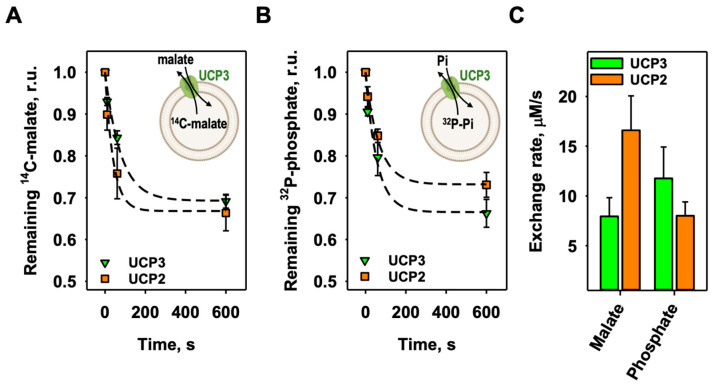
The homoexchange of malate and phosphate is different for mUCP3 and mUCP2. (**A**) Efflux of malate from liposomes over time measured by remaining intraliposomal ^14^C-malate radioactivity of mUCP3 (green triangles) or mUCP2 (orange squares). Concentration of intraliposomal and extraliposomal malate was 2 mM. Inset: Experimental setup to test malate/malate homoexchange for mUCP3 (or mUCP2). (**B**) Efflux of phosphate from liposomes over time measured by remaining intraliposomal ^32^P-phosphate radioactivity of mUCP3 (green triangles) or mUCP2 (orange squares). Concentration of intraliposomal and extraliposomal phosphate was 2 mM. Inset: Experimental setup to test phosphate/phosphate homoexchange for mUCP3 (or mUCP2). Lines in (**A**,**B**) are a least-squares fit of an exponential function to the data. (**C**) Exchange rate calculated for the malate/malate exchange (first bar set) and the phosphate/phosphate exchange (second bar set) of mUCP3 (green) and mUCP2 (orange). Membranes were made of 45:45:10 mol% PC:PE:CL. The buffer solution contained 50 mM Na_2_SO_4_, 10 mM Tris, 10 mM MES, and 0.6 mM EGTA at pH = 7.34 and T = 296 K. Lipid concentration was 4 mg/mL; protein concentration was 4 µg/(mg lipid). The data are the mean ± SD of at least three independent experiments.

**Figure 4 biomolecules-14-00021-f004:**
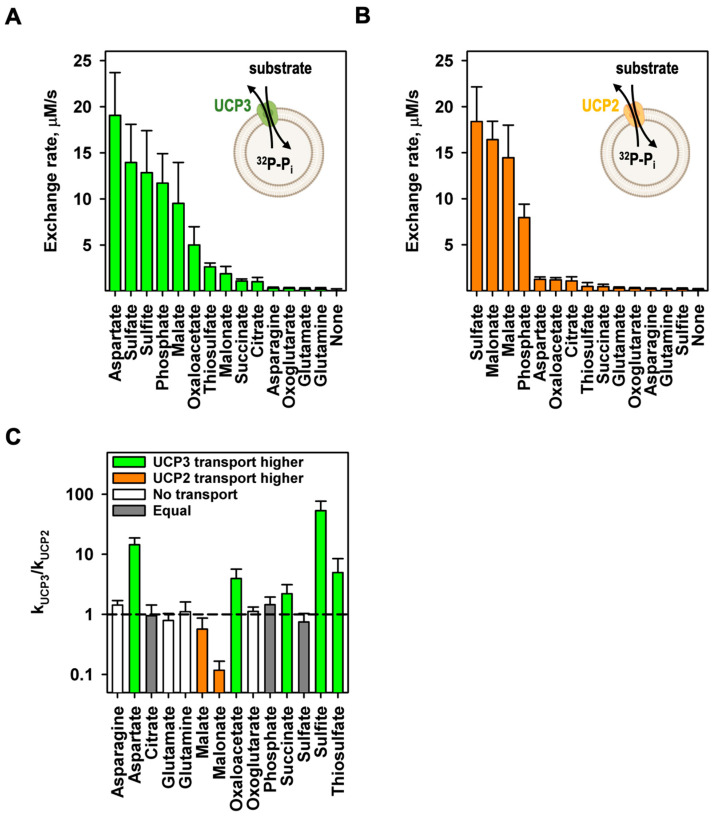
UCP3 catalyzes the heteroexchange of different substrates against phosphate. Exchange rates of mUCP3 (**A**) and mUCP2 (**B**) of the exchange of intraliposomal phosphate against a set of different external substrates sorted by their exchange rate. “None” displays the uniport activity of UCP3 and UCP2 measured in the absence of any external substrate. The time course of phosphate efflux for all measurements is shown in Appendix A. (**C**) Ratio of the mUCP3-mediated exchange rates to the mUCP2-mediated exchange rates of the substrates sorted by alphabet. Green bars indicate higher transport rates for the substrates by mUCP3, and orange bars for mUCP2. Grey bars indicate similar transport rates for the substrates by mUCP3 and mUCP2. White bars indicate substrates that are not transported by mUCP3 or mUCP2. Experimental conditions are indicated in Appendix A.

**Figure 5 biomolecules-14-00021-f005:**
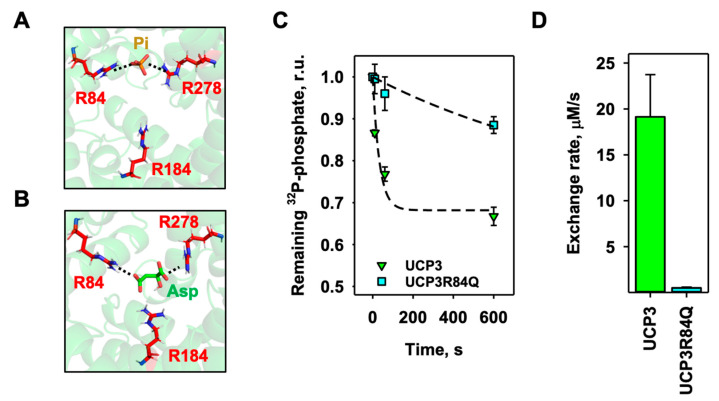
R84 plays an essential role in UCP3-mediated substrate exchange. (**A**,**B**) Static docking image of possible interactions of (**A**) phosphate (P_i_) with R84 and R184 and (**B**) aspartate (Asp) with R84 and R278 in the central binding site of mUCP3. Substrates are displayed as licorice in orange (**A**) and green (**B**), the amino acids as licorice in red, and mUCP3 in cartoon representation in green. (**C**) Time course of aspartate/phosphate exchange over time of mUCP3 (green) and mUCP3R84Q mutants (cyan), measured by remaining ^32^P-phosphate radioactivity inside proteoliposomes. The concentration of intraliposomal phosphate and extraliposomal aspartate was 2 mM. (**D**) Exchange rate calculated for the aspartate/phosphate exchange of mUCP3 (left) and mUCP3R84Q (right). In the experiments, lipid concentration was 4 mg/mL and protein concentration was 4 µg/(mg lipid). Membranes were made of PC:PE:CL (45:45:10 mol%) and liposomes filled with 2 mM ^32^P-phosphate. The buffer solution contained 50 mM Na_2_SO_4_, 10 mM Tris, 10 mM MES, and 0.6 mM EGTA at pH = 7.34 and T = 296 K. The data are the mean ± SD of at least three independent experiments.

**Table 1 biomolecules-14-00021-t001:** Summary of exchange rates reported in this paper.

**1. Protein Specificity of the Malate/Phosphate Exchange**					
**Substrates**		**Exchange Rate, µM/s**			
		**UCP3**	**UCP2**	**ANT1**	**None**
Phosphate	Malate	9.57 ± 4.29	14.5 ± 3.5	0.19 ± 0.01	0.23 ± 0.09
**2. Substrate specificity of UCP3 and UCP2**					
		**Exchange rate,** **µM/s**		**Exchange rate** **mmol/(min ⋅ g_Protein_)**	
		**UCP3**	**UCP2**	**UCP3**	**UCP2**
*Homoexchange*					
Malate	Malate	7.94 ± 1.88	16.6 ± 3.5	9.93 ± 2.35	20.8 ± 4.3
Phosphate	Phosphate	11.7 ± 3.20	8.00 ± 1.40	14.7 ± 4.0	10.0 ± 1.8
*Heteroexchange*					
Phosphate	Asparagine	0.40 ± 0.04	0.27 ± 0.04	0.50 ± 0.05	0.34 ± 0.05
	Aspartate	19.1 ± 4.6	1.31 ± 0.19	23.9 ± 5.8	1.64 ± 0.24
	Citrate	1.10 ± 0.39	1.14 ± 0.38	1.38 ± 0.49	1.43 ± 0.48
	Glutamate	0.29 ± 0.07	0.36 ± 0.06	0.36 ± 0.08	0.45 ± 0.08
	Glutamine	0.27 ± 0.12	0.24 ± 0.02	0.34 ± 0.15	0.30 ± 0.03
	Malate	9.57 ± 4.39	14.5 ± 3.50	12.0 ± 5.5	18.1 ± 4.4
	Malonate	1.96 ± 0.72	16.5 ± 2.0	2.45 ± 0.90	20.6 ± 2.5
	None	0.13 ± 0.12	0.18 ± 0.06	0.16 ± 0.15	0.23 ± 0.08
	Oxaloacetate	5.07 ± 1.92	1.26 ± 0.18	6.34 ± 2.40	1.58 ± 0.23
	Oxoglutarate	0.37 ± 0.04	0.32 ± 0.04	0.46 ± 0.05	0.40 ± 0.05
	Succinate	1.16 ± 0.17	0.52 ± 0.19	1.45 ± 0.21	0.65 ± 0.24
	Sulfate	14.0 ± 4.1	18.4 ± 3.7	17.5 ± 5.1	23.0 ± 4.7
	Sulfite	12.9 ± 4.5	0.24 ± 0.06	16.1 ± 5.6	0.30 ± 0.08
	Thiosulfate	2.70 ± 0.36	0.54 ± 0.36	3.38 ± 0.45	0.68 ± 0.45

## Data Availability

The datasets generated and/or analyzed during the current study can be obtained upon reasonable request from the corresponding authors.

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
