# Peer review of "Uncoupling Protein 3 Catalyzes the Exchange of C4 Metabolites Similar to UCP2"

_biomolecules, 2023, doi:10.3390/biom14010021_

Round 1

Reviewer 1 Report

Comments and Suggestions for Authors

The manuscript “Uncoupling protein 3 catalyzes the exchange of C4 metabolites similar to UCP2” (biomolecules-2664646) by Kreiter et al. describes the transport activity of isolated mouse UCP3. The approach used is sound and the results demonstrate that UCP3 transports similar substrates to UCP2. The major concerns of this reviewer regard the lack of experiments showing if UCP3 has any characteristics different from its closest biochemically-characterized homologs UCP2, UCP4, UCP5 and UCP6. Therefore, major revision is required before accpetance for publishing.

Major comments

Whereas the most important substrates of UCP2 were tested on UCP3 in this study, the UCP5 and UCP6 substrates sulfite and thiosulfate were not. These substrates should be tested on UCP3.

The authors should test if GTP (and GDP) is an inhibitor of UCP3 in their transport assay because it has been tested for UCP2.

UCP4 has been suggested to catalyze uniport transport of aspartate previously. Can it be excluded that UCP3 catalyzes uniport transport of phosphate and aspartate to a certain extent, leaking radioactive substrate in the 10 min transport experiment or on the 30 ml size exclusion resin column (lines 138-139) following this incubation? The graphs for the heteroexchanges in Fig. 4A-B with “None” external substrate should be displayed in Supplementary Figs. S4 and S5.

Fig. 5 C. The phosphate and aspartate heteroexchange transport experiment presented with the UCP3R84Q mutant suggests that the phosphate or aspartate transport may be affected. The transport experiments should also be performed with phosphate and aspartate homoexchange to show if both aspartate and phosphate transport are affected or only the transport of one of these substrates.

Fig. 5 A-B. To show that R184 and R278 are really involved in phosphate and aspartate binding, respectively, these UCP3 mutants should be generated and evaluated with transport experiments with phosphate and aspartate homoexchange.

Minor comments

Lines 105-112. The ingredients of the TE/G buffer should be defined.

Line 109. GTP was added in the reconstitution buffer, presumably as a stabilizing inhibitor. Are the authors sure that no GTP remains in the transport experiments after dialysis?

Line 173. The docking performed with AutoDock is described as “static”. In AutoDock residues may be chosen in the grid box to have flexible side chains for a “semi-rigid” docking. If this option was used it should be described better (which flexible residues were chosen) in the Methods. If this option was not used it should be used.

Lines 192-196 and elsewhere in the text and figures. It should be preferred to use “mmol/min/g protein” for transport rates and not “microM/s”.

The illustrations of proteoliposomes in the insets of fig. 3 A-B. For clarity the line with the double direction arrows should be substituted with two lines with arrows in opposite directions as in Fig. 2 A because at least in the initial time points the radioactively labelled substrate is likely to go out of the liposomes and the cold substrate in if UCP3 catalyzes antiport transport.

In Fig. 4 and supplementary material “oxogluterate” should be changed to “oxoglutarate”.

Fig. 5 A-B. It seems remarkable that phosphate is bound by R184 whereas aspartate is bound by R278. In the the figure it looks like the two arginines are on the same transmembrane helix in the figure. The figures should illustrate the observed differences in binding of phosphate and aspartate better, and the atom types of aspartate should be displayed with different colours.

Lines 318-321. Transport rates in different conditions cannot be directly compared, but maybe the relative rates between different substrates. If the authors want to compare transport kinetics they should determine the Km values for phosphate and aspartate.

Reviewer 2 Report

Comments and Suggestions for Authors

In this MS, the authors studied the exchange rates of several metabolites through mitochondrial uncoupling protein 3, as compared to uncoupling protein 2, measured in a model system of unilamellar liposomes. The main findings are a certain similarity of transported substrated including phosphate and succinate, and the more UCP3-specific substrate aspartate.

The MS is well written, experiments appear to have been expertly conducted, and conclusions are in line with presented results. This report is of wide interest in the field of mitochondrial physiology. I recommend acceptance after resolving the following minor issues:

p.1, l.36: In this context, the term “uncoupling” is generally used to refer to uncoupling of oxygen consumption from ATP production, not “proton gradient from ATP production”. I suggest to rephrase to avoid confusion.

p.2, l. 49: “aerobic glycolysis” => this is either a typo and should be “anaerobic”, or the authors must explain the concept referred to.

p.3, Figure 3, legend: Please explain the meaning of the motif labelled in light blue in the figure legend.

entire MS: I recommend adding more explanations that may appear obvious for the authors and other experts in this methodology, but will greatly assist understanding for many among the broad readership of this journal:

-          Explain directionality in this assay, both in terms of facing of inserted proteins, and substrate flux. Explain how that translates into insight into the directionality of transport across the inner mitochondrial membrane, or lack thereof.

-          Provide some context on the concept of homoexchange. Is this expected? Which solute carriers are known to exhibit this feature? What is the assumed driving force?

-          Contextualize the determined transport rates. How do these compare with transport rates for other solute exchangers of the mitochondrial inner membrane, e.g. the OGC, CIC, DIC etc etc.?

Round 2

Reviewer 1 Report

Comments and Suggestions for Authors

To make the the charatcterization of the UCP3 transport function complete, I still think it is necessary that:

1) the substrates suggested (sulfite and thiosulfate) should be tested as substrates of UCP3 (point 1 of the previous review report)

2) GTP (and GDP) should be tested as inhibitors of UCP3 (point 2 of the previous review report)

3) uniport transport of aspartate catalyzed by UCP3 should be tested (point 3 of the previous review report).
